# Systematic Review of Roles of Arecoline and Arecoline *N*-Oxide in Oral Cancer and Strategies to Block Carcinogenesis

**DOI:** 10.3390/cells12081208

**Published:** 2023-04-21

**Authors:** Albert Min-Shan Ko, Hung-Pin Tu, Ying-Chin Ko

**Affiliations:** 1Department of Biomedical Sciences, Graduate Institute of Biomedical Sciences, Chang Gung University, Taoyuan 33302, Taiwan; 2Cardiovascular Department, Chang Gung Memorial Hospital, Taoyuan 33302, Taiwan; 3Health Aging Research Center, Chang Gung University, Taoyuan 33302, Taiwan; 4Department of Public Health and Environmental Medicine, School of Medicine, College of Medicine, Kaohsiung Medical University, Kaohsiung 80756, Taiwan; 5Department of Medical Research, China Medical University Hospital, China Medical University, No. 2 Yu-Der Road, Taichung 40447, Taiwan; 6Graduate Institute of Toxicology, College of Medicine, National Taiwan University, Taipei 106216, Taiwan

**Keywords:** arecoline, arecoline *N*-oxide, areca nut, betel quid, carcinogenesis, oral cancer, oral submucous fibrosis, epithelial-mesenchymal transition, epigenetics

## Abstract

Betel quid and areca nut are complex mixture carcinogens, but little is known about whether their derived single-agent arecoline or arecoline *N*-oxide (ANO) is carcinogenic, and the underlying mechanisms remain unclear. In this systematic review, we analyzed recent studies on the roles of arecoline and ANO in cancer and strategies to block carcinogenesis. In the oral cavity, flavin-containing monooxygenase 3 oxidizes arecoline to ANO, and both alkaloids conjugate with *N*-acetylcysteine to form mercapturic acid compounds, which are excreted in urine, reducing arecoline and ANO toxicity. However, detoxification may not be complete. Arecoline and ANO upregulated protein expression in oral cancer tissue from areca nut users compared to expression levels in adjacent normal tissue, suggesting a causal relationship between these compounds and oral cancer. Sublingual fibrosis, hyperplasia, and oral leukoplakia were diagnosed in mice subjected to oral mucosal smearing of ANO. ANO is more cytotoxic and genotoxic than arecoline. During carcinogenesis and metastasis, these compounds increase the expression of epithelial–mesenchymal transition (EMT) inducers such as reactive oxygen species, transforming growth factor-β1, Notch receptor-1, and inflammatory cytokines, and they activate EMT-related proteins. Arecoline-induced epigenetic markers such as sirtuin-1 hypermethylation, low protein expression of miR-22, and miR-886-3-p accelerate oral cancer progression. Antioxidants and targeted inhibitors of the EMT inducers used reduce the risk of oral cancer development and progression. Our review findings substantiate the association of arecoline and ANO with oral cancer. Both of these single compounds are likely carcinogenic to humans, and their mechanisms and pathways of carcinogenesis are useful indicators for cancer therapy and prognosis.

## 1. Introduction

Areca nut (AN) is the fruit of *Areca catechu*, which grows in the tropical Pacific, South Asia, Southeast Asia, and mainland China. For thousands of years, AN has been regarded as safe. This nut is consumed with tobacco (South and Southeast Asia) or without tobacco in mainland China (AN husk only), Taiwan (5% of all users consume inflorescences), the Solomon Islands (AN seeds), and Papua New Guinea. Currently, an estimated number of 600 million individuals consume betel quid (BQ) worldwide [1]. BQ contains slaked lime and other additives. AN is also commonly used in BQ prepared with or without tobacco. Although the Internal Agency for Research on Cancer (IARC) acknowledged in 1985 that tobacco-containing BQ is carcinogenic to humans (Group 1: sufficient evidence in humans; causal relationship established), the question of whether tobacco-free BQ is carcinogenic remains unclear [2]. Most relevant epidemiological studies have been conducted in India and adjacent regions, where only 5% of all users consume tobacco-free BQ [3], which hinders the sampling of eligible candidates for analyzing the effects of tobacco-free BQ consumption. Ko et al. [4] reported that approximately 2 million individuals consume tobacco-free BQ in Taiwan. Subsequently, they revealed that the odds of developing oral cancer were 28 times higher in BQ users (tobacco-free) than in BQ nonusers after adjusting for education and occupation covariates [5]. The incidence of oral cancer was computed to be 123-fold higher in patients who smoked, drank alcohol, and chewed betel quid than in abstainers [5], indicating their interaction. IARC reported that excluding AN, no other BQ ingredients and their metabolites, betel leaf, or lime are associated with cancer [6,7]. A AN has about 10% crude fiber, but no research has shown that physical or mechanical wear in the oral mucosa is associated with cancer [6,7]. On the basis of independent studies conducted in Taiwan [4], Pakistan [8], and India [9], in 2004, the IARC reassessed the causal relationship between BQ use and oral cancer and concluded that both BQ and AN without tobacco are Group 1 carcinogens [6]. In 2012, the IARC further reported that BQ without added tobacco causes cancer of the oral cavity and esophagus. Some epidemiological studies have evaluated cancers in other sites, but the data are insufficient for drawing conclusions [7]. The agency acknowledged the existence of sufficient evidence for the carcinogenicity of AN in experimental animals (mice or rats) receiving AN through gavage or subcutaneous injection. However, animal models have exhibited increased incidences of lung cancer, hepatocellular carcinoma, and fibrosarcoma, but not oral cancer. AN use may lead to cancer of the oral cavity, esophagus, and pharyngeal squamous cells, which are the oral site of cancer initiation, but not cancer elsewhere. Therefore, studies must focus on clarifying the carcinogenic effects of AN and its ingredients on the oral cavity and upper aero-digestive tract, particularly oral and esophageal squamous cells [10].

Arecoline, arecaidine, guvacoline, and guvacine are the major alkaloids found in AN; arecoline has been demonstrated to induce mutagenicity, genotoxicity, and cytotoxicity in various experimental models [10,11]. Baseline salivary arecoline levels (>0.1 µg/mL) were detectable in 69% (22/32) of all BQ users; the maximum level increased from 6 to 97 µg/mL after AN consumption [11]. In an experimental study, the mean highest concentration of arecoline was found to be 44 µg/mL after 5 min of chewing 5 g of AN, and this concentration decreased to 3 µg/mL by 35 min [12]. Venkatesh et al. also suggested that the lower levels of salivary arecoline (>0.1 µg/mL) achieved during and after chewing a low dose of AN (0.5 g) to be enough to cause collagen and cytotoxic effects on oral tissues over a period of time in chronic chewers [13]. Arecoline may also be metabolized into arecoline *N*-oxide in the oral cavity. Using LTQ Orbitrap spectrometry, salivary arecoline *N*-oxide adduct-bound annexin A1 was detected in AN-treated human keratinocytes [14]. In 2020, arecoline was classified as possibly carcinogenic to humans (Group 2B: limited evidence in humans; less than sufficient evidence in experimental animals) on the basis of strong mechanistic evidence [15]; however, no human data were available from the IARC. Limited information is available about arecoline *N*-oxide. Giri et al. [16] demonstrated the oxidation of arecoline into arecoline *N*-oxide by human flavin-containing monooxygenase (FMO)1 and FMO3. Arecoline *N*-oxide exerted stronger mutagenic effects on bacteria than arecoline [17], and may be a carcinogen. The oxidation of trimethylamine into trimethylamine *N*-oxide is catalyzed by FMO3 produced by gut microbiota [18]; this reaction has been associated with colorectal cancer [19]. Likewise, arecoline *N*-oxide may be involved in oral carcinogenesis. In the present systematic review, we searched for and synthesized recent studies providing evidence for the carcinogenicity of arecoline and arecoline *N*-oxide and the strategies available to block carcinogenesis.

## 2. Materials and Methods

### 2.1. Search Strategy

In October 2022, the PubMed (National Center for Biotechnology Information) and Web of Science databases were searched for relevant studies; the updated Preferred Reporting Items for Systematic Review [20] guidelines were followed in the present study. The following English string was used: (arecoline), (arecoline *N*-oxide). Articles were retrieved from these databases that had been published between January 2015 and October 2022. All articles were downloaded to Endnote software X9 and filtered by their title, after which the title and abstract were reviewed. Filters in the aforementioned databases were used to remove review articles, letters, and comments from the search results. We have registered the methods section of the manuscript in OSF Registries (Registration DOI: https://doi.org/10.17605/OSF.IO/SGP5D (accessed on 24 February 2023)).

### 2.2. Study Selection and Data Extraction

As shown in Figure 1, 603 records were initially obtained from the databases. After removing duplicate records and non-text articles, 359 records remained. Their abstracts were reviewed to assess their eligibility: 309 articles were excluded, whereas 50 were reviewed. A total of 36 studies were excluded subsequently because they were conducted using simple cellular models. Finally, 14 studies published in the past 8 years met the inclusion criterion. This criterion was providing relevant evidence for the carcinogenicity of arecoline and arecoline *N*-oxide by using patient tissues, normal human and cancer cells, and animal models. Explanations and comments are provided on the basis of all 14 articles.

## 3. Results

### 3.1. Arecoline N-Oxide

Kuo et al. [14] showed that immunodeficiency mice received arecoline *N*-oxide (500 µg/mL) for 21 weeks, which resulted in increased collagen deposition and severe squamous hyperplasia with elevated expression of γ-H2AX protein (DNA damage marker) in the sublingual tissues of the mice. In normal human gingival fibroblasts, the protein-adduct of arecoline *N*-oxide bound to annexin A1 was identified in saliva. Arecoline and arecoline *N*-oxide induce cytotoxicity, increase the levels of collagen type I and oral-fibrosis-related proteins (e.g., transforming growth factor (TGF)-β1, S100 calcium-binding protein A4 (S100A4), matrix metalloproteinase (MMP)-9, interleukin (IL)-6, fibronectin, and α-smooth muscle actin (SMA)), and reduce the level of E-cadherin in normal human gingival fibroblasts. These compounds further increase the level of 8-hydroxy-2′-deoxyguanosine, another marker of oxidative stress and DNA damage, which indicates their high potential to induce oral potentially malignant disorders (OPMDs).

Chang et al. [21] reported that the treatment of immunodeficiency and healthy mice with 500 and 1500 µg/mL arecoline *N*-oxide, respectively, increased the expression levels of caspase-8 and γ-H2AX in the sublingual hyperplastic lesions of the mice; moreover, substantially elevated (compared with the levels in healthy oral tissues) expression levels of proliferation proteins (Ki67 and proliferating cell nuclear antigen (PCNA)) were observed in the hyperplastic tissues. In Kuo’s subsequent study [22], the researchers treated three groups of healthy mice (through mucosal treatment) with arecoline, arecoline *N*-oxide, or dimethyl sulfoxide (control), and they extended the observation period from 20 to 26 weeks. Arecoline and arecoline *N*-oxide induced oral leukoplakia in the sublingual tissues of 40% and 78% of the treated mice, respectively. The expression levels of Ki67 and PCNA were upregulated in the nucleus of hyperplastic lesions of the mice treated with the test compounds. Furthermore, elevated expression levels of Notch receptor-1 (NOTCH1), HES1, and FAT1 were noted in the hyperplastic lesions of the experimental mice. Arecoline tended to increase FMO3 levels in HGF1 cells, whereas arecoline *N*-oxide decreased these levels, which suggests that FMO3 is involved in arecoline oxidation.

The aforementioned research team collected 22 pairs of oral squamous cell carcinoma (OSCC) tissues and adjacent normal oral mucosa specimens from patients with oral cancer (>82% were BQ users) [23]. Compared with the healthy tissues, the tumor tissues exhibited significantly elevated concentrations of arecoline and arecoline *N*-oxide and expression levels of a somatic protein (NOTCH1) and proinflammatory markers (IL-1β and IL-17). In OC-3 oral carcinoma cells, which were cultured from only an AN user, arecoline *N*-oxide led to higher levels of cytotoxicity and intracellular reactive oxygen species (ROS) than did the other metabolites of arecoline. *N*-acetylcysteine was significantly downregulated (approximately 40%) in cells treated with arecoline and arecoline *N*-oxide, whereas glutathione was downregulated (70–80%) in cells treated with arecoline and its metabolites. Arecoline *N*-oxide significantly increased the expression levels of proteins involved in the mitogen-activated protein kinase (MAPK) pathway; cytokines such as IL-1β, IL-6, IL-8, and IL-17a; and chemokines such as C–C motif chemokine ligand (CCL)2, CCL5, granulocyte-colony stimulating factor (G-CSF), and nuclear factor (NF)-κB.

In a study conducted by Wang et al. [24] using hepatocytes, arecoline *N*-oxide exhibited stronger cytotoxicity, caused more DNA damage, and exhibited greater mutagenicity than did arecoline. Antioxidants such as *N*-acetylcysteine, Trolox, and penicillamine protected hepatocytes against arecoline-*N*-oxide-induced DNA damage and ROS production.

### 3.2. Arecoline

In a mouse study conducted by Wen et al. [25], the experimental group received 1000 mg/L arecoline through drinking water and the control group received distilled water for 20 weeks. In week 8, an increase was noted in the expression level of collagen type I in the lamina propria of the tongue, and this level was even higher in week 20. Mouth opening gradually decreased from 10.9 to 8.2 mm in the experimental group but increased from 10.6 to 14.6 mm in the control group.

Using the OSCC tissues of 68 AN users, Hu et al. [26] analyzed S100A4 expression through immunohistochemistry. S100A4 expression was found to be positively correlated with clinical grade, lymph-node metastasis, and poor patient survival. In oral epithelial cells, the downregulation of S100A4 markedly reversed the outcomes of arecoline treatment. LY294002 (phosphatidylinositol-3 kinase (PI3K) inhibitor), SP600125 (c-Jun *N*-terminal kinase inhibitor), and CAY10585 (hypoxia-inducible factor-1α inhibitor) were found to inhibit arecoline-induced S100A4 expression in oral epithelial cells.

Ho et al. [27] reported that arecoline treatment led to a dose-dependent increase in the expression of zinc finger E-box-binding homeobox-1 (ZEB1) in normal oral epithelial cells (SG and FaDu cells). The downregulation of ZEB1 considerably reversed the effects of arecoline-induced carcinogenesis, including cell migration and invasiveness. In OSCC, ZEB1 is more highly expressed in recurrent lesions than in primary lesions.

Chuerduangphui et al. [28] reported that a low concentration of arecoline (0.025 µg/mL) increased the viability and proliferation of OSCC cells and induced the expression of IL-6, signal transducer and activator of transcription 3 (STAT3), and cellular myelocytomatosis oncogene (c-Myc). In arecoline-treated OSCC cells, the expression of mir-22 was suppressed. By contrast, oncostatin M expression was substantially upregulated; this was inversely correlated with miR-22 expression. Thus, arecoline may upregulate oncostatin M expression by inducing c-Myc expression and reducing miR-22 expression.

Zhang et al. [29] demonstrated that the expression level of miR-886-3p in OSCC cells was negatively correlated with the concentration of arecoline. Compared with the levels in adjacent tissues, the expression level of miR-886-3p was reduced in OSCC tissues. miR-886-3p transfection resulted in reduced viability, migration, and invasion of OSCC cells. Thus, arecoline-mediated inhibition of miR-886-3p may be involved in OSCC proliferation and metastasis.

Islam et al. [30] demonstrated that sirtuin-1 (SIRT1) was considerably hypermethylated in OSCC tissues obtained from AN users and nonusers compared with the level of methylation in the oral mucosa of healthy controls. Compared with the findings in the control group, the methylation level of SIRT1 was considerably increased and the expression levels of SIRT1 mRNA and protein were markedly decreased in human oral cells treated with arecoline (50 µg/mL). Thus, DNA hypermethylation may be an early event (occurring before observable clinical changes) in the development of oral cancer.

Ren et al. [31] showed that arecoline (160 µg/mL) successfully induced the epithelial–mesenchymal transition (EMT) and enhanced the migration and invasion of CAL33 and UM2 OSCC cells. The levels of several inflammatory cytokines—such as serum amyloid A1 (SAA1), IL-6, IL-36G, and chemokines such as CCL2 and CCL20—were considerably altered in the arecoline-induced EMT. In an orthotopic xenograft model of oral cancer established using BALB/c nude mice, arecoline enhanced cervical lymph node metastasis; this suggests that arecoline promotes the metastasis of oral cancer cells in vivo.

In a study conducted by Uehara et al. [32], human gingival epithelial progenitor cells were treated with arecoline (50 µg/mL) for 3 days and then with no arecoline for 3 days (alternating treatment with no treatment for a total period of 30 days). MMP-9 expression was found to be markedly upregulated in the experimental group compared with the expression in the control group. The inhibitors of NF-κB/IκB, MAPK, p38 MAPK, and STAT3 decreased the arecoline-induced expression level of MMP-9, which implies that the associated signaling pathways are involved in MMP-9 production.

Using the database from the GSE139869 and the Cancer Genome Atlas—OSCC, data were mined for the identification of the differentially expressed genes to explore metastatic disease, which remains the primary cause of death in patients with OSCC, especially those who use AN. Li et al. found six (PLAU, IL1A, SPP1, CCL11, TERT, and COL1A2) of thirteen genes in association with the arecoline, which related to cuproptosis and might play an important role in the metastasis of OSCC [33].

## 4. Discussion

The oral mucosa comprises an epithelium and the lamina propria; keratinocytes are found in the epithelium, and fibroblasts are found in the lamina propria. Several OPMDs (e.g., oral leukoplakia, oral submucous fibrosis, and erythroplakia) can affect the lamina propria. The most common type of oral epithelial cancer is OSCC. The aforementioned OPMDs may share regulatory mechanisms and pathways involved in carcinogenesis. Human normal oral cells, such as keratinocytes and fibroblasts, are used as cellular models to generate mechanistic evidence; dysplasia keratinocytes and OSCC tissues and cells are used to study pathological changes. Moreover, saliva contains several important factors that can be evaluated in the present context. Of the 14 articles selected in this systematic review, 12 investigated the roles (in vivo and in vitro) of arecoline and arecoline *N*-oxide in various human normal oral and OSCC cells and tissues or animal experiments.

### 4.1. Arecoline and Its Metabolites in the Oral Cavity

In the oral cavity, arecoline (Mass 156) or arecoline *N*-oxide (Mass 172) conjugate with *N*-acetylcysteine (Mass 163) and form mercapturic acid compounds (Mass 319 or 335), which are excreted through urine, reducing the toxicity of arecoline and arecoline *N*-oxide. Conjugation is the second stage of standard detoxification, in which the toxicity of toxic compounds is reduced through urinary excretion. This phenomenon represents the natural response of the human body to xenobiotics [10]. Figure 2 presents the molecular weight and structure of the aforementioned mercapturic acid compounds. However, the detoxification may be incomplete, and arecoline or arecoline *N*-oxide are still present and involved in carcinogenesis.

Kuo et al. [22] demonstrated that in normal human gingival fibroblasts, arecoline increased and arecoline *N*-oxide decreased the levels of FMO3, which indicates that FMO3 is depleted during arecoline metabolism in the oral cavity. Furthermore, the expression levels of FMO3 and γ-H2AX were reduced in arecoline-treated OSCC cells, which highlights the possible involvement of FMO3 in arecoline-induced DNA damage: FMO3 functions as a metabolic enzyme for the production of arecoline *N*-oxide in oral cells.

Arecoline, arecoline *N*-oxide, and cysteine are found in saliva. Arecoline induces human gingival fibroblast and oral cancer cells to produce FMO3. *N*-acetylcysteine and arecoline are found in AN-induced oral cancer cells. Therefore, the oral cavity and adjacent tissues appear to produce arecoline, arecoline *N*-oxide, and their metabolites, which play key roles in carcinogenesis.

### 4.2. Association of Arecoline N-Oxide with OSCC

Studies conducted using human tissues, animals, and cellular models to explore human carcinogens have varied on the basis of cell type, dose, and treatment duration; the results must be reproducible. With the advancement of relevant techniques, the definition has recently been included in mechanistic studies to complement the evidence generated using human and animal data. Table 1 summarizes a total of six studies investigating the association of arecoline *N*-oxide with OSCC.

#### 4.2.1. Cellular Models

Arecoline *N*-oxide increases levels of cytotoxicity, genotoxicity, ROS, proinflammatory cytokines, and chemokines; it can also affect the levels of proteins involved in MAPK pathways. In normal human gingival fibroblasts, arecoline *N*-oxide treatment increased the expression levels of TGF-β1, S100A4, IL-6, fibronectin, and α-SMA but decreased that of E-cadherin [14]; the expression levels of IL-1ß, tumor necrosis factor-α, and NOTCH1 were also increased [22]. In OSCC cells obtained from AN users, arecoline *N*-oxide treatment increased the expression levels of IL-1β, IL-6, CCL2, G-CSF, NF-κB, NOTCH1, and proteins involved in the MAPK pathways [23]. In mice and human models, arecoline *N*-oxide increased the levels of NOTCH1 protein levels in oral hyperplasia tissues. ROS [34], NOTCH1 [35], TGF-β1, and various cytokines and chemokines [36] are regarded as major EMT inducers. The EMT is involved in the cancer development and metastasis, inflammation and fibrotic disorders. Various mesenchymal-specific macromolecules––including α-SMA, collagen type 1, fibronectin, vimentin, *N*-cadherin, and S100A4––are associated with the EMT. MMP-9 expression is inconsistent with arecoline *N*-oxide. Further studies are needed to confirm the response of MMP-9 expression to arecoline *N*-oxide exposure. Cellular models indicate the role of arecoline *N*-oxide in carcinogenesis through EMT inducers such as ROS, TGF-β1, NOTCH1, inflammatory cytokines, and chemokines, and then activate EMT proteins.

The results of the aforementioned studies conducted using human, animal, or cellular models confirm the carcinogenicity of arecoline *N*-oxide.

#### 4.2.2. Animal Models

In an animal study, 6-week-old C57BL/6 mice treated with 500 µg/mL of arecoline for 28 weeks were found to develop no oral lesions or tumors [37]. Kuo et al. [14] employed an oral mucosa smear of arecoline *N*-oxide to mimic the manner in which humans chew BQ; arecoline *N*-oxide (starting dose, 500 µg/mL; duration, 21 weeks) was administered to 10-week-old immunodeficient mice (weight, approximately 27 g). Sublingual fibrosis, hyperplasia, cytotoxicity, and genotoxicity were observed in the mice. In healthy mice, the dose of arecoline *N*-oxide was increased to 1500 µg/mL for 26 weeks. In addition to developing hyperplasia, 78% of the mice developed sublingual leukoplakia. A slight increase was noted in their bodyweight. The dose or duration was not increased further to validate the results, possibly because in the later stage of the experiment, mouth opening was substantially reduced, making the smear difficult. Establishing mouse models of oral cancer through oral mucosa smearing is a challenging task. Five decades ago, Suri et al. [38] induced oral cancer within 3 weeks by applying AN to the cheeks of hamsters. Notably, their results could not be replicated even when the same protocols were followed [39]. In subsequent animal studies [37,40,41,42], standard carcinogens, such as 7,12-dimethylbenz[a]anthracene and 4-nitroquinoline 1-oxide, were required to develop cancer. Arecoline *N*-oxide, which is highly toxic, can help establish OPMDs in animal models relatively easily, although oral mucosal administration may not induce tumor formation. In future animal studies, tumors could be induced through subcutaneous injection and gavage; in addition, species that are larger than mice can be used.

#### 4.2.3. Human Models

The role of arecoline *N*-oxide in carcinogenesis has been investigated in a series of relevant studies [14,21,22,23]. The protein expression of arecoline *N*-oxide markedly increased (two-fold), and the higher expression levels of IL-1β, IL-17, and NOTCH1 in OSCC tissues compared with the levels in adjacent normal tissues. Using clinical samples, these paired case–control studies provided direct evidence for a causal relationship between the test compounds and oral cancer. Carcinogens can be directly detected in the tissues of patients with cancer. Ghosh et al. [43] reported that tissue samples obtained from patients with bladder cancer exhibited elevated levels of inorganic arsenic, possibly resulting from the consumption of drinking water containing excess arsenic. In a human study, Nithiyanantham et al. [23] reported that arecoline *N*-oxide was found in OSCC tissues, indicating patients with oral cancer were still consuming BQ immediately before cancer tissues were collected from them for pathological examination.

### 4.3. Association of Arecoline with OSCC

Three recent studies have indicated the involvement of the EMT in arecoline-induced carcinogenesis. EMT-related proteins (e.g., S100A4, ZEB1, and SAA1) were detected in arecoline-treated OSCC cells and patient tissues. ZEB1 is involved in EMT induction, and SAA1 promotes the EMT-mediated migration and invasion of OSCC cells. Notably, S100A4 was detected in human gingival fibroblasts treated with arecoline and arecoline *N*-oxides; higher expression levels of S100A4 [44] and ZEB1 [45] were found in oral submucous fibrosis tissues and buccal mucosal fibroblasts, respectively, which indicated the development of oral submucous fibrosis from normal mucosal fibroblasts and the progression to cancer and metastasis. In normal gingival cells, oral dysplasia, and cancer patient specimens, arecoline has been demonstrated to induce the expression of MMP-9 [32,46], but not in keratinocytes and OSSC cells [47]. Heavy metal ions are essential micronutrients, but either insufficient or excessive abundance of metals can trigger cell death [48]. In a cuproptosis study, Li et al. [33] showed that there is a close association between arecoline, cuproptosis, and cancer-associated fibroblasts. Using the bioinformatics approach, six cuproptosis-related genes showed that their area under the curve (AUC) values were 0.6; this seems to be a marginal significance, and needs further study to increase the AUC values and verify the cellular mechanism.

In a recent mouse study, Wen et al. [25] showed that the oral administration of arecoline (1000 µg/mL) for 20 weeks (through drinking water) did not result in any pathological changes in the oral mucosa. Subcutaneous injection of arecoline may lead to cancer but not oral cancer.

Nithiyanantham et al. [23] demonstrated higher levels (four-fold) of arecoline-induced protein expression in the oral cancer tissues of AN users than in their adjacent normal tissues. Arecoline increased the levels of cytotoxicity, genotoxicity, proinflammatory cytokines, chemokines, and MAPK pathway proteins in both normal and OSCC cells obtained from AN users and nonusers. Elevated expression levels of arecoline-induced proteins in OSCC tissues obtained from clinical samples of BQ/AN users prove the carcinogenicity of arecoline in humans.

Epigenetics also promote the levels of EMT proteins in arecoline-induced carcinogenesis. Epigenetic changes mainly include DNA methylation, histone modification, and noncoding RNA and microRNA syntheses. These changes are reversible because they affect the expression of genes without altering their DNA sequence. Three recent studies have implicated epigenetics in arecoline-induced carcinogenesis. OSCC cells were stimulated with different doses of arecoline; a low dose induced the expression of miR-22, whereas a high dose induced the expression of miR-886-3p; the expression of both microRNAs was downregulated in OSCC cells. These microRNAs are tumor suppressors and are important in arecoline-induced carcinogenesis. Elevated levels of DNA methylation increase cancer risk. SIRT1 is a tumor suppressor. SIRT1 hypermethylation has been observed in OSCC cells and the tissues of AN users; thus, SIRT1 hypermethylation may be a biomarker of oral cancer.

### 4.4. Association of Arecoline Addiction with OSCC

In a series of studies conducted by the Asian Betel-Quid Consortium in Mainland China, Taiwan, Malaysia, Indonesia, Nepal, and Sri Lanka, the prevalence of BQ abuse and dependence was discovered to be 1–46% and 3–39%, respectively, and BQ addiction has been associated with OPMDs [49,50,51]. The same investigation revealed that moderate to severe BQ use disorder was associated with OPMD risk [52]. BQ chewers without BQ use disorder had a 5.7-fold higher risk of OSCC than did non-chewers, and individuals with severe BQ use disorder had a 42-fold higher risk of OSCC than did non-chewers [53]. Several strategies are needed to reduce BQ dependence. High levels of BQ use result in genetic variations in the monoamine oxidase A (MAOA) gene; arecoline inhibits the mRNA and protein expression of MAOA [54]. MAOA inhibitors such as arecoline prevent the breakdown of neurotransmitters and increase the levels of dopamine and serotonin in the brain, which are associated with the brain’s reward, cognitive, and impulsive systems [55]. The long-term use of AN may lead to AN dependence, which increases the risks of OPMD and OSCC. An epidemiological study revealed that the use of antidepressants [e.g., MAOA inhibitors and selective serotonin reuptake inhibitors (SSRIs)] is associated with reduced risk of oral cancer occurrence [56]. In an animal study, mice were trained to freely drink AN-containing water; the administration of MAOA inhibitors and SSRIs reduced the consumption of AN-containing water, thus reducing the incidence of oral fibrosis in mice [57]. The same research group observed a significant reduction in BQ use in patients receiving antidepressants compared with the level of BQ use in patients with untreated depressive symptoms [58]. In a randomized, double-blind, placebo-controlled trial [59], fixed low doses of an MAOA inhibitor (moclobemide) and an SSRI (escitalopram) for 8 weeks conferred therapeutic benefits (5.6-fold) in patients with BQ use disorder.

### 4.5. Targeted Inhibitors Reduce the Carcinogenesis of Arecoline and Arecoline N-Oxide

#### 4.5.1. Antioxidants: *N*-Acetylcysteine and Glutathione

Arecoline or arecoline *N*-oxide can increase the levels of intracellular ROS in different cell lines [60,61,62]. High levels of ROS were detected in the saliva of BQ users [60,63]. Using the dichloro-dihydro-fluorescein diacetate fluorescence method, arecoline, arecoline *N*-oxide, and their mercapturic acid compounds were demonstrated to increase, to varying degrees, the levels of intracellular ROS in oral cancer cells obtained from BQ users [23]. The effects of arecoline *N*-oxide were stronger than those of its acetylated metabolites. Arecoline *N*-oxide downregulated the expression of antioxidant enzymes, such as *N*-acetylcysteine and glutathione. By contrast, arecoline *N*-oxide mercapturic acid upregulated the expression of *N*-acetylcysteine and glutathione [23]. *N*-acetylcysteine and glutathione may exert dual effects on arecoline and arecoline *N*-oxide: one may exhibit antioxidant activities, whereas the other may facilitate xenobiotic detoxification. The formation of mercapturic acid compounds leads to an increase in the levels of antioxidant enzymes and a decrease in those of intracellular ROS, thus reducing the risk of cancer.

#### 4.5.2. Inhibitors of NOTCH1 and Anti-Inflammatory Agents

The NOTCH1 signaling pathway has been investigated as a therapeutic target for the treatment of cancers and inflammatory disorders. The most common of these are γ-secretase inhibitors, but the application of different NOTCH1 inhibitor modalities faces many challenges in the preclinical and early clinical stages [64,65]. A number of approaches to inhibit NOTCH1 have potential benefits in various clinical aspects of cancer. Downregulation of NOTCH1 can be used alone or in combination with chemotherapy. Inhibitors targeting TGF-ß have been considered by pharmaceutical companies for cancer therapy, some of which are cautiously undergoing clinical trials [66]. Several drugs that block IL-1 are currently on the market for the treatment of rheumatic diseases [67]. In addition, several anti-inflammatory drugs, such as celecoxib and curcumin, may be available for BQ-associated cancers. However, all of these off-label uses require clinical trials to assess their efficacy and safety [10].

### 4.6. Association of Arecoline and Arecoline N-Oxide with Increased Risks of other Cancers

The IARC [7] reported that the use of BQ/AN even without added tobacco causes cancer of the oral cavity and esophagus. Although several epidemiological studies have explored the risk of BQ/AN use–related cancer in other sites, the obtained data are insufficient to draw conclusions. A positive association between BQ use without added tobacco and liver cancer has been observed in six studies; however, the analysis was not adjusted for smoking and alcohol use [7]. Arecoline and arecoline *N*-oxide may be metabolized in the liver. In hepatocytes, arecoline *N*-oxide exhibited higher cytotoxicity and caused more DNA damage than did arecoline [24]. A study conducted using human hepatoma cells revealed that low concentrations of arecoline induced cell proliferation and migration by activating the PI3K/Akt/mammalian target of rapamycin (mTOR) pathway [68]; unfortunately, these findings were not validated in studies using human, animal, or cellular models. Notably, an epidemiological study also indicated that BQ use was not associated with gastric or colorectal cancer [69]. In contrast, BQ-associated cancers involving squamous cells of the oral cavity, pharynx, larynx, and esophagus have been reported [9,69], showing cell-specificity.

### 4.7. N-Nitrosated Metabolites of Arecoline

Wenke and Hoffmann [70] reported that the *N*-nitrosation of arecoline leads to the formation of *N*-nitrosoguvacoline (NGL), 3-methylnitrosamino propionitrile (MNPN), and 3-methylnitrosamino propionaldehyde (MNPA). In an earlier study, MNPN was not detected in the saliva of current BQ users in Taiwan [71]; by contrast, Prokopczyk et al. [72] reported the presence of MNPN in the saliva of current BQ users living in New York. MNPN has been demonstrated to induce cancer in animal models [73]. The carcinogenicity of MNPN has been established using experimental animals. The IARC has classified MNPN as a Group 2B human carcinogen and NGL, *N*-nitrosoguvacine, and MNPA as Group 3 carcinogens (not classifiable as human carcinogens) [6].

MNPN is reportedly derived from arecoline; both compounds are possibly carcinogenic to humans (Group 2B), and they can be compared in toxicity, particularly for a toxic dose. Sundqvist et al. [74] reported that a low concentration of arecoline exerted stronger cytotoxic and genotoxic effects than did that of MNPN. At a concentration of 1.6 mM, arecoline reduced the colony formation efficiency (survival) of human buccal epithelial cells by 50%; by contrast, at concentrations of up to 5 mM, MNPN exerted no prominent effects on cell survival. At a concentration of 5 mM, arecoline reduces DNA single-strand breaks by 0.5 per 10 Da × 10 Da, whereas the same concentration of MNPN reduces DNA single-strand breaks by only 0.2 per 10 Da × 10 Da. In an earlier study, the median concentration of MNPN in saliva after AN chewing was 1.5 (range, 0.5–11.5) ppb [72], which is too low to induce cancer; conversely, the concentration of arecoline in saliva has been reported to be 44–140 ppm in other studies [11,12,75]. Unfortunately, since the publication of the aforementioned study, no studies have reported MNPN in saliva or urine. Future studies should include cellular and animal models to evaluate the carcinogenicity of relatively low doses of MNPN.

### 4.8. Pathways Involved in Carcinogenesis Induced by Arecoline and Arecoline N-Oxide

The pathways involved in carcinogenesis are highly complex; the underlying mechanisms remain to be elucidated. As shown in Figure 3, carcinogenesis induced by arecoline and arecoline *N*-oxide comprises the following six steps from cancer initiation and cancer progression: BQ/AN use disorder or addiction; arecoline and arecoline *N*-oxide formation in oral cavity; EMT inducers’ activation; ROS formation, cytotoxicity, genotoxicity, and inflammation; genetic and epigenetic pathway dysfunction; and the EMT induced cancer and metastasis. OSCC mainly occurs in epithelium while OPMDs mainly occurs in lamia proper, and some of them may turn into cancer. 

The evidence strongly suggests an association between arecoline addiction and oral cancer [51,52,53]. In a recent randomised clinical trial, Hung et al. [59] recruited 111 male AN use disorders by the Diagnostic and Statistical Manual of Mental Disorders (DSM-5-defined; age mean SD, 41.3 ± 9.5 year; duration of AN use more than 24 years), gave fixed low doses of moclobemide and escitalopram for 8 weeks, and conferred therapeutic benefits (5.6-fold) in patients with AN use disorder. The underlying mechanisms (such as brain MAOA, serotonin, and reward system) have also been clarified to some extent. The human body treats arecoline as a xenobiotic compound; detoxification of arecoline results in the formation of its more toxic metabolite arecoline *N*-oxide. The compounds are metabolized into mercapturic acids to reduce the cancer risk. This mechanism is similar to that underlying alcohol detoxification. Alcohol dehydrogenase metabolizes alcohol into acetaldehyde, a highly toxic substance and known carcinogen, which is then metabolized into acetate; acetate is broken down into water and nontoxic carbon dioxide [76]. The present study improves our understanding of the pathways involved in the carcinogenesis induced by arecoline and arecoline *N*-oxide; EMT inducers such as ROS, TGF-β1, NOTCH1, IL-1, and IL-6 promote the EMT pathways (such as fibronectin, S100A4, SMA, MMP-9, and E-cadherin) and lead to the occurrence of abnormal chromosomes (5, 20, 21) in OSCC [77].

## 5. Conclusions

Studies conducted using human, animal, and cellular models have generated evidence on the carcinogenicity of arecoline and arecoline *N*-oxide in humans. These compounds increase the levels of protein expression in the oral cancer tissues of AN users compared with the levels in adjacent normal tissues. By contrast, in animal models, administration of arecoline and arecoline *N*-oxide through the oral mucosa rarely leads to the development of cancer; other methods such as subcutaneous injection or gavage may lead to cancers, excluding oral cancer. Arecoline *N*-oxide has also been demonstrated to be more cytotoxic and genotoxic than arecoline. Both compounds conjugate with *N*-acetylcysteine to produce mercapturic acid compounds, which are relatively less toxic and are excreted through urine. Furthermore, both arecoline and arecoline *N*-oxide increase the expression levels of EMT inducers, such as ROS, NOTCH1, and cytokines, and decrease epigenetic protein expression, activating the EMT pathway, and ultimately leading to the development of cancer and its progression to a metastatic stage. These carcinogenesis mechanisms and pathways need replication study to confirm their associations; carcinogenesis blockers also need clinical trials to verify their efficacy.

In a randomized clinical trial, fixed low doses of MAOA inhibitor (moclobemide) and SSRI (escitalopram) for 8 weeks conferred therapeutic benefits in patients with AN use disorder. Arecoline functions as a MAOA inhibitor. AN use stimulates the reward, cognition, and impulsivity centers of the brain, and thus leads to AN dependence and cancer. Current antidepressants such as MAOA inhibitors and SSRIs can effectively prevent AN use and avoid cancer risk. However, this study provides preliminary evidence and requires replication in larger trials. *N*-acetylcysteine or other nutritional supplements exhibit antioxidant properties; their long-term use may reduce cancer risk. Current evidence improves our understanding of the association of arecoline and arecoline *N*-oxide with oral cancer. A better understanding of the detailed mechanisms and pathways by which single compounds induce cancer is a potential target for therapeutic development.

## Figures and Tables

**Figure 1 cells-12-01208-f001:**
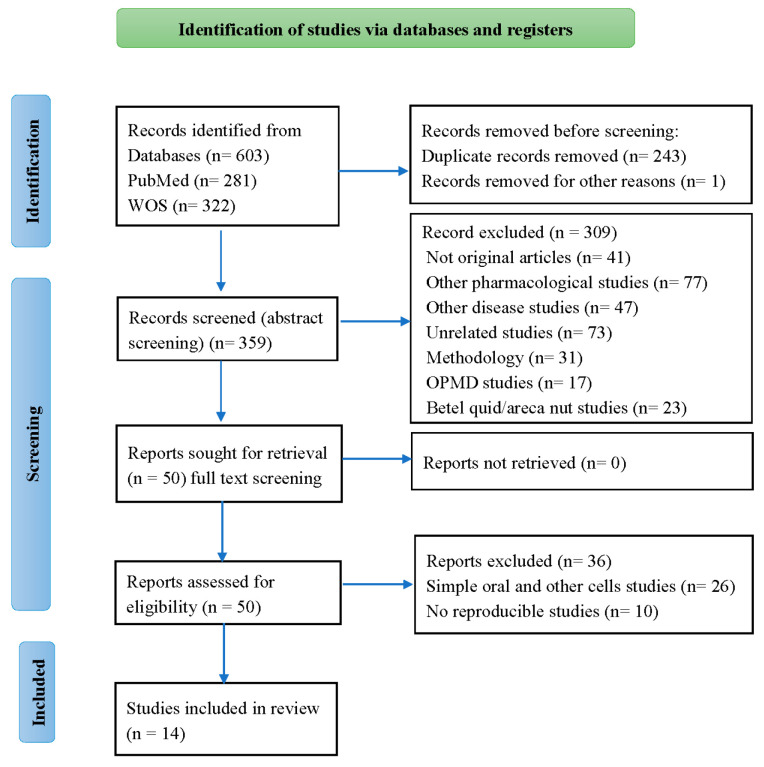
Systematic review flowchart. Finally, evidence, explanations, and comments are provided on the basis of all 14 articles.

**Figure 2 cells-12-01208-f002:**
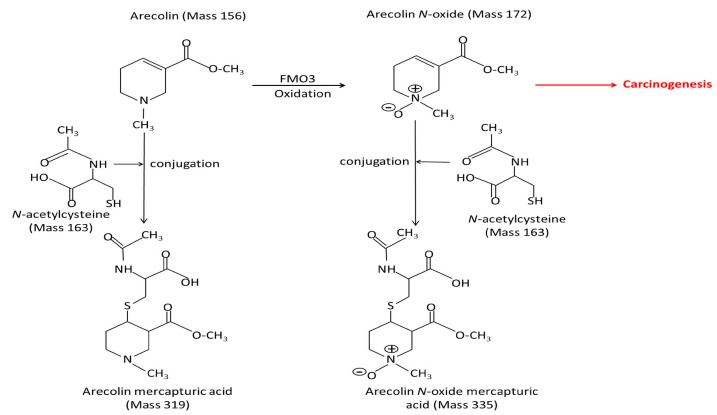
FMO3 oxidizes arecoline into arecoline *N*-oxide in the oral cavity. According to molecular weight and structure, both alkaloids may be carcinogenic and conjugate with *N*-acetylcysteine to form mercapturic acid compounds, which are then removed through urinary excretion, reducing toxicity. However, they supply *N*-acetylcysteine, which seems to help block carcinogenesis. Nonetheless, the detoxification may not be complete, and arecoline or arecoline *N*-oxide is still present and involved in carcinogenesis.

**Figure 3 cells-12-01208-f003:**
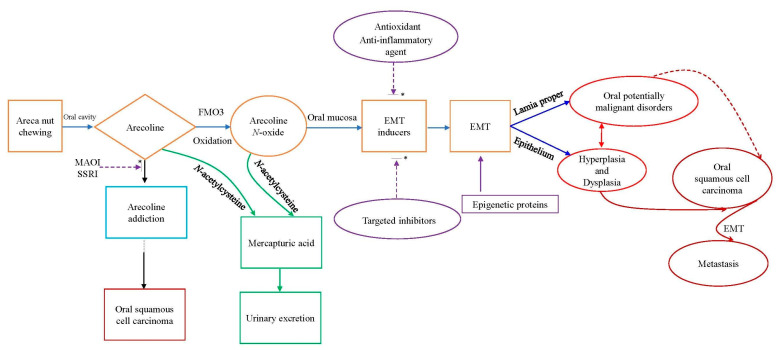
Pathway showing the roles of arecoline and arecoline *N*-oxide in carcinogenesis. Six steps from cancer initiation and cancer progression: BQ/AN use or addiction; arecoline and arecoline *N*-oxide formation; EMT inducers activation: ROS formation, cytotoxicity, genotoxicity, and inflammation; genetic and epigenetic pathway dysfunction and the EMT induced cancer and metastasis. The pathways sometimes lead to OPMDs before the development of cancer. EMT, epithelial-mesenchymal transition; MAOI: monoamine oxidase inhibitor; and SSRI, selective serotonin reuptake inhibitors. EMT inducers: reactive oxygen species, transforming growth factor beta-1, Notch receptor-1, interleukin-1beta, and interleukin-6. EMT markers: E-cadherine. S100 calcium-binding protein A4, zinc finger E-box-binding homeobox-1, and alpha smooth muscle actin. Epigenetics: sirtuin-1 hypermethylation, decrease in miR-22, miR-886-3p. *—Blocking effect.

**Table 1 cells-12-01208-t001:** Role of arecoline *N*-oxide (ANO) in carcinogenesis.

Authors (Reference)	Study Models	Carcinogenesis Effect	Association/Mechanism
Lin et al., 2011c [17]	Bacteria strains (TA98, TA100) tester	Mutagenicity; Arecoline: weak; ANO: moderate	*N*-acetylcysteine, glutathione, and cysteine inhibit mutagenicity
Kuo et al., 2015 [14]	Fibroblasts; Keratinocytes; NOD/SCID mice	DNA damage, Collagen↑, Hyperplasia, Cytotoxicity↑	EMT inducers:TGF-β1↑; EMT↑: S100A4, IL-6, MMP-9, α-SMA, Fibronectin; E-cadherin↓
Chang et al., 2017 [21]	Oral cancer patients; Dysplasia keratinocytes; NOD/SCID mice	DNA damage, Proliferation, Hyperplasia	Caspase-8↑
Kuo et al., 2015 [22]	C57BL/6 mice; Dysplasia keratinocytes; Fibroblasts; OSCC cells (HSC-3, SCC-9)	Collagen↑, Hyperplasia, Leukoplakia, Proliferation, DNA damage	EMT inducers: NOTCH1↑; NOTCH1↓ (Knockdown NOTCH in OSCC cells); IL-1β↑; TNF-α↑; FAT1↑; P53↓
Nithiyanantham et al., 2021 [23]	Paired case–control study (22 pairs cancer tissue); NOD/SCID mice; OSCC cells (HSC3, SAS, OC-3-derived from areca nut users)	ANO↑ in cancer tissues; Cytotoxicity↑; DNA damage	EMT inducers: ROS↑; L-1β↑, IL-6↑, IL-8↑, IL-17↑, CCL2↑, CCL5↑, G-CSF↑, NF-κB↑; Glutathione↓; *N*-acetylcysteine↓; MAPK pathway (ERK, JNK, P38)
Wang et al., 2018 [24]	Hepatocytes; Bacteria strains (TA98, TA100) tester	DNA damage; Cytotoxicity↑; Mutagenicity↑	EMT inducers: ROS↑; *N*-acetylcysteine, Trolox, Penicillamine inhibit DNA damage

NOD/SCID mice, nonobese diabetic severe-combined immunodeficiency mice; EMT, epithelial–mesenchymal transition; OSCC, oral squamous cell carcinoma; and ROS, reactive oxygen species. ↑ Increased; ↓ Reduced.

## Data Availability

Not applicable.

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
