# Peer review of "Systematic Review of Roles of Arecoline and Arecoline N-Oxide in Oral Cancer and Strategies to Block Carcinogenesis"

_cells, 2023, doi:10.3390/cells12081208_

Round 1

Reviewer 1 Report

- Line 45: what do you mean by "Group 1"?

- In the discussion section, the authors should start discussing studies carried out on the cellular models, followed by the ones done on the animal models and finally the studies performed on human models.

- The section "4-8. Pathways involved in carcinogenesis induced by arecoline and arecoline N-oxide" needs to be more developed.

- Figure 3 should be moved to the paragraph 4-8.

- Line 429: Based on two or three studies to suggest the association between arecoline addiction and oral cancer is insignificant. Other factors are to be taken into account, namely the history of the diseases of the patients participating in the trials, the lifestyle, the hygiene..... As a result, these parameters should be considered in this systematic review.

- Conclusion requires refinement by resuming the preeminent deductions of the studies and builds-up recommendations for more assessment and clinical trials to be conducted to understand clearly the connection between arecoline, arecoline N-oxide and oral cancer.

Author Response

Reviewer 1

Comments and Suggestions for Authors

- Line 45: what do you mean by "Group 1"?

Authors’ Response:

We appreciated the reviewer for careful reading our manuscript and revised it.

Thank you very much for your helpful and insightful comments.

We have added the information (lines 47-48; Group 1: sufficient evidence in humans; causal relationship established).

- In the discussion section, the authors should start discussing studies carried out on the cellular models, followed by the ones done on the animal models and finally the studies performed on human models.

Authors’ Response:

Thank you very much for your comments.

We have followed your suggestion and have moved the section of 4-2-1. Cellular models, 4-2-2. Animal models, and 4-2-3. Human models.

- The section "4-8. Pathways involved in carcinogenesis induced by arecoline and arecoline N-oxide" needs to be more developed.

Authors’ Response:

Thank you very much for your comments.

We have added some statements.

- Figure 3 should be moved to the paragraph 4-8.

Authors’ Response:

Thank you very much for your comments.

We have moved to the sections 4-8.

- Line 429: Based on two or three studies to suggest the association between arecoline addiction and oral cancer is insignificant. Other factors are to be taken into account, namely the history of the diseases of the patients participating in the trials, the lifestyle, the hygiene..... As a result, these parameters should be considered in this systematic review.

Authors’ Response:

Thank you very much for your comments.

We have followed your suggestion and added the information in sections 4-8.

- Conclusion requires refinement by resuming the preeminent deductions of the studies and builds-up recommendations for more assessment and clinical trials to be conducted to understand clearly the connection between arecoline, arecoline N-oxide, and oral cancer.

Authors’ Response:

Thank you very much for your comments.

We have followed your suggestion and added the information in the Conclusion section.

Reviewer 2 Report

The manuscript is acceptable to be publish in Cells after minor revision. The Fig.'s 1,2 and 3 should be presented in more readable form.

Author Response

Reviewer 2

Review Report (Round 1)

Comments and Suggestions for Authors

The manuscript is acceptable to be publish in Cells after minor revision. The Fig.'s 1,2 and 3 should be presented in more readable form.

Authors’ Response:

We appreciated the reviewer for careful reading our manuscript and revised it.

Thank you very much for your helpful and insightful comments.

We have added the information for Fig.'s 1,2 and 3 and they would be readable.

Reviewer 3 Report

It is a well-organized review by ko et al. The authors summarized the studies really well. 

I. Author must comment on association of Arecoline and ANO and cuproptosis. 

II. Comment from  study of salivary arecoline in areca nut chewers Lippincott Williams & Wilkins https://journals.lww.com  by D Venkatesh (2018) can be added  

Author Response

Reviewer 3

Review Report (Round 1)

Comments and Suggestions for Authors

It is a well-organized review by ko et al. The authors summarized the studies really well.

Authors’ Response:

We appreciated the reviewer for careful reading our manuscript and revised it.

Thank you very much for your helpful and insightful comments.

  1. Author must comment on association of Arecoline and ANO and cuproptosis.

Authors’ Response:

Thank you very much for your helpful and insightful comments.

We have followed your suggestion and added 2 Cuproptosis papers (references 33 and 48) and comments have been given.

  1. Comment from study of salivary arecoline in areca nut chewers Lippincott Williams & Wilkins https://journals.lww.com by D Venkatesh (2018) can be added

Thank you very much for your comments.

We have followed your suggestion and added 1 salivary arecoline paper (reference 13).

Reviewer 4 Report

The research direction of this manuscript belongs to the minority, however the review is still valuable. Some suggestions to better serve readers:

1. In the introduction, the authors described the ways in which betel nut is used by people in different regions, for example by mixing it with tobacco. However, there was no subsequent description of the interaction between the components in the betel nut and those in the tobacco leaf. Please complete or delete unnecessary descriptions.

2. The canceration caused by mechanical wear of the coarse fiber of areca nut should also be properly described.

3. Out of respect for the author, the author's name should be included in the citation instead of in one study, for example in line 106 and 117.

4. I think the publication period and keywords can be appropriately expanded.

5. The summary of the work in the literature is insufficient.

6. Figure 3 needs to be embellished

7.In line 212, the exclusive description needs to be rigorous.

8. The conclusion is too short, can be appropriate outlook.

Author Response

Reviewer 4

Review Report (Round 1)

Comments and Suggestions for Authors

The research direction of this manuscript belongs to the minority, however the review is still valuable. Some suggestions to better serve readers:

Authors’ Response:

We appreciated the reviewer for careful reading our manuscript and revised it.

  1. In the introduction, the authors described the ways in which betel nut is used by people in different regions, for example by mixing it with tobacco. However, there was no subsequent description of the interaction between the components in the betel nut and those in the tobacco leaf. Please complete or delete unnecessary descriptions.

Authors’ Response:

Thank you very much for your comments.

We have followed your suggestion and added the information in Introduction section (lines 52-59).

  1. The canceration caused by mechanical wear of the coarse fiber of areca nut should also be properly described.

Authors’ Response:

Thank you very much for your comments.

We have followed your suggestion and added the information in the Introduction section (lines 57-59).

  1. Out of respect for the author, the author's name should be included in the citation instead of “in one study”, for example in lines 106 and 117.

Authors’ Response:

Thank you very much for your comments.

We have added the information of the author's name in all text.

  1. I think the publication period and keywords can be appropriately expanded.

Authors’ Response:

Thank you very much for your comments.

We have expanded one year and a paper added one article for review (reference 33).

We have added keywords.

  1. The summary of the work in the literature is insufficient.

Authors’ Response:

Thank you very much for your comments.

We have followed your suggestion and added the information in the Conclusion section.

  1. Figure 3 needs to be embellished

Authors’ Response:

Thank you very much for your comments.

We have reorganized it in Figure 3.

7.In line 212, the exclusive description needs to be rigorous.

Authors’ Response:

Thank you very much for your comments.

We have deleted the sentences in sections 4-1.

  1. The conclusion is too short, can be appropriate outlook.

Authors’ Response:

Thank you very much for your comments.

We have followed your suggestion and added the information in the Conclusion section.

Round 2

Reviewer 1 Report

- Put the "N" in the compound's name in Italic font: N-oxide, arecoline N-oxide, N-acetylcysteine, N-nitrosated,....etc.

- Line 463: Remove the empty space between the scheme and the title for the Figure 3.

- Check the font style for the references highlighted in red.

Author Response

Author's Reply to the Review Report (Reviewer 1)

Comments and Suggestions for Authors

Authors’ Response:

Thank you very much for your helpful and insightful comments.

- Put the "N" in the compound's name in the Italic font: N-oxide, arecoline N-oxide, N-acetylcysteine, N-nitrosated,....etc.

Authors’ Response:

We sincerely appreciate your constructive advice.

We have followed your suggestion and have put the "N" in the compound's name in an Italic font in this revised manuscript (including text, graphical abstract, Figures 2 and 3, and Table 1).

- Line 463: Remove the empty space between the scheme and the title for the Figure 3.

Authors’ Response:

Thank you very much for your comments.

We have removed the empty space between the scheme and the title for Figure 3.

- Check the font style for the references highlighted in red.

Authors’ Response:

Thank you very much for your comments.

We have corrected the font style for references 13, 33, 48, and 77.

Reviewer 3 Report

THe revised manuscript can be accepted for publication in "CELLS" journal.

Author Response

Author's Reply to the Review Report (Reviewer 3)

Comments and Suggestions for Authors

The revised manuscript can be accepted for publication in "CELLS" journal.

Authors’ Response:

Thank you very much for your helpful and insightful comments.

Reviewer 4 Report

I have no further questions.

Author Response

Author's Reply to the Review Report (Reviewer 4)

Comments and Suggestions for Authors

I have no further questions.

Authors’ Response:

Thank you very much for your helpful and insightful comments.